# Effects of Audiovisual Interactions on Working Memory Task Performance—Interference or Facilitation

**DOI:** 10.3390/brainsci12070886

**Published:** 2022-07-06

**Authors:** Yang He, Zhihua Guo, Xinlu Wang, Kewei Sun, Xinxin Lin, Xiuchao Wang, Fengzhan Li, Yaning Guo, Tingwei Feng, Junpeng Zhang, Congchong Li, Wenqing Tian, Xufeng Liu, Shengjun Wu

**Affiliations:** 1Department of Military Medical Psychology, Air Force Medical University, Xi’an 710032, China; hy6548182902022@163.com (Y.H.); zhguo1016@163.com (Z.G.); wxl961010@163.com (X.W.); xlxsunkewei@126.com (K.S.); lxx199205@163.com (X.L.); wangxiuchao1984@163.com (X.W.); psyfengzhan@fmmu.edu.cn (F.L.); 13659205403@163.com (Y.G.); ftw_fmmu@163.com (T.F.); zjp0505@163.com (J.Z.); lllmmu@fmmu.edu.cn (X.L.); 2School of Public Health, Shaanxi University of Chinese Medicine, Xianyang 712046, China; congchong1011@163.com (C.L.); twqvenchi@163.com (W.T.)

**Keywords:** audiovisual interaction, working memory, cognitive load, interference effect, visual dominance, speed-accuracy trade-off

## Abstract

(1) Background: The combined n-back + Go/NoGo paradigm was used to investigate whether audiovisual interactions interfere with or facilitate WM. (2) Methods: College students were randomly assigned to perform the working memory task based on either a single (visual or auditory) or dual (audiovisual) stimulus. Reaction times, accuracy, and WM performance were compared across the two groups to investigate effects of audiovisual interactions. (3) Results: With low cognitive load (2-back), auditory stimuli had no effect on visual working memory, whereas visual stimuli had a small effect on auditory working memory. With high cognitive load (3-back), auditory stimuli interfered (large effect size) with visual WM, and visual stimuli interfered (medium effect size) with auditory WM. (4) Conclusions: Audiovisual effects on WM follow the resource competition theory, and the cognitive load of a visual stimulus is dominated by competition; vision always interferes with audition, and audition conditionally interferes with vision. With increased visual cognitive load, competitive effects of audiovisual interactions were more obvious than those with auditory stimuli. Compared with visual stimuli, audiovisual stimuli showed significant interference only when visual cognitive load was high. With low visual cognitive load, the two stimulus components neither facilitated nor interfered with the other in accordance with a speed–accuracy trade-off.

## 1. Introduction

### 1.1. Study of Working Memory

Since Baddeley proposed the concept of working memory in 1974, it has been widely recognized by researchers [1]. In the past 20 years, working memory (WM) research has become a hot topic and frontier in cognitive neuroscience. As an essential component of executive function, working memory is essential for the function of the brain and refers to a memory system with limited capacity for temporarily storing and managing information during the performance of cognitive tasks [2,3]. Among many working memory models, Baddeley’s multicomponent model is relatively mature and well-established and has attracted widespread attention from researchers [4]. He suggested that working memory consists of two main systems: the storage system (which includes the phonological loop and visuospatial sketchpad) and the processing system. The phonological loop is responsible for the storage of auditory-based information; the visuospatial sketchpad is responsible for the storage of visual-based information. The main processing system is the central executive (CE), which is responsible for the coordination of subsystem functions in working memory, the control of encoding and extraction strategies, the operation of the attention management system, and the extraction of information from long-term memory [5]. Many studies in recent years have shown that working memory, as the core of human cognitive activity [6], is closely related to higher cognitive activities in humans, such as intellectual activities, logical reasoning, mathematical abilities, and problem solving [7,8,9,10], mainly due to the central executive system playing its role [11,12,13]. On the one hand, the central executive system regulates the activity of attention, and on the other hand, it controls behavioral responses. Due to its specificity, it is considered both the core component of working memory and the most advanced and complex component of working memory [2,14,15]. Through structural equation modelling and neuroimaging of the central executive system, early researchers [16,17] found that the central executive system includes three subfunctions, namely, updating, switching, and inhibitory functions, which are closely related to individuals’ fluid intelligence and academic performance [18]. Specifically, the updating function refers to the process of monitoring and encoding newly encountered information and constantly modifying information irrelevant to the current operation based on the original stored information, thereby assimilating the new information in place of the old information [19]. The classical research paradigms of the updating function include the n-back task, letter memory task and keep track task [20]. Inhibitory function refers to the ability to consciously inhibit automatic and dominant responses, especially the ability to keep information irrelevant to the current target task from entering working memory [21]. Researchers have found that almost all executive function tasks involve inhibition, which is a core component of the central executive system of working memory, to ensure for the smooth functioning of the execution system in working memory, as well as switching, which is needed to suppress automated processing, and updating, which is needed to eliminate information that is not useful [15]. In recent years, researchers have determined that the extent of an individual’s inhibitory capacity is mainly related to cognitive load, that is, the higher the cognitive load, the worse the individual’s inhibitory capacity [22]. Furthermore, as a factor of the task itself, the cognitive load in the task will affect not only the accuracy of working memory but also the interference effect of external distractions [23]. On the one hand, according to the discrete-fixed precision model, the capacity of an individual’s working memory is limited, and the accuracy of information preservation will decrease monotonically with an increase in the cognitive load of the task [24]. On the other hand, according to attentional load theory, the interference effect will increase with an increase in cognitive load [25], i.e., under low cognitive load, although both the target stimulus and interference stimulus can be perceived, the interference effect can be suppressed by goal-oriented attentional control. However, under high cognitive load, as there are no more cognitive resources to maintain top-down processing, it is difficult to suppress the interference effect, i.e., working memory capacity deteriorates, indirectly indicating a decrease in working memory accuracy and an increase in cognitive load [26]. The classical research paradigms of the inhibitory function include the Go/NoGo task, Stroop task, and stop-signal task [20]. The switching function is also known as the conversion of attention or the conversion of task, which refers to flexible switching between two or more tasks, operations or thinking modes that occurs simultaneously according to the requirements of the task instruction language. The classical research paradigms of the switching function include the local-global task and number–letter task [27]. Meanwhile, many studies have found that working memory impairments are associated with schizophrenia [28,29], attention-deficit/hyperactivity disorder (ADHD) [30,31], Alzheimer’s disease (AD) [32], and autism spectrum disorders (ASD) [33]. Since working memory is closely related to people’s lives and learning, it plays a crucial role in people’s lives. Therefore, the study of working memory has become a multidisciplinary focus.

### 1.2. Study of Audiovisual Interactions

In the information age, we learn and remember information through interactions between different sensory modalities rather than being limited to a single sensory channel. In daily life, a large amount of sensory information is often present simultaneously (e.g., visual, auditory, sensory, olfactory, etc.), so the brain must concurrently process information from each sensory channel. That is, our brain integrates stimulus information from multiple sensory channels to control behavioral responses and engage in higher cognitive activities, such as attention and memory, to guide us through our normal lives [34]. However, our central system has a limited capacity to process information; therefore, to accomplish different cognitive tasks and improve our brain’s efficiency in processing information, we must selectively attend to the information content delivered by specific sensory channels. Previous studies have shown that, when multiple sensory channels are processed simultaneously, visual stimuli can not only adjust the perception of auditory stimuli, but also affect the perception of tactile stimuli [35], proprioception [36], and somatosensation [37]. The processing advantage of this visual pathway over other sensory pathways is called the Visual dominance effect. Visual dominance occurs not only during perception but also during rapid response tasks, such as the Colavita effect, the phenomenon of preferentially responding to visual stimuli when both visual and auditory stimuli are presented simultaneously [38]. It has been reported that 94% of the information enters the brain through visual and auditory channels, making them the two most important channels through which humans receive information, and they play an important role in higher human cognitive processing [39,40,41].

There is research on the effects of audiovisual interactions on higher cognitive functions in humans. The most classic example is the classic McGurk effect proposed by McGurk and MacDonald in Nature in 1976 [42]. The McGurk effect can be observed in an experiment in which an audiovisual clip is provided in which a person repeats the sound “ba” while a person’s mouth movement associated with “ga” is shown on the screen. During the experiment, the visual stimulus and the auditory stimulus with their distinct pronunciation appeared simultaneously. If the subjects looked at the mouth shape with their eyes while listening to the repeating sound with their ears, the subjects reported hearing a “da” sound; however, if the screen showed a person pronouncing “ba” on the screen, subjects reported hearing the “ba” sound. If the subjects closed their eyes and listened to the voice over without looking at the shape of the mouth, the subjects also reported hearing the “ba” sound [43]. The McGurk effect illustrates the importance of the “ba” sound and illustrates that our vision has an effect on hearing. For example, when visual and auditory stimuli are presented simultaneously and the content is inconsistent, the receiver will deviate from sound recognition, i.e., there is interference between the visual and auditory channels, and cognitive bias occurs [44]. When visual and auditory stimuli are presented simultaneously and the content is consistent, the visual and auditory channel information will facilitate each other. Diaconescu et al. [45] found that subjects recognized semantically congruent stimuli presented in dual channels more quickly than stimuli presented in single channels. The introduction of the McGurk effect set off a wave of research on stimulus patterns investigating audiovisual interactions. In addition to visual stimuli interfering with the acquisition of auditory information, numerous studies have found that auditory stimuli can affect visual perception, which is an audiovisual interaction. Shams et al. found through behavioral and electroencephalography (EEG) studies that, when auditory stimuli were added to visual experiments, the reporting of the number of flicker stimuli was influenced by the number of short tones. For example, when two short pure tones close together were interspersed with a flash of light, the subjects were prone to the visual hallucinations and judged one flash as two, which is the well-known flash illusion effect [46,47,48,49]. Later, Aderson et al. [50] found that, when the audiovisual stimuli were congruent, the accuracy of subjects’ judgments were higher when one flash was accompanied by one pure tone or two flashes were accompanied by two pure tones, and reaction times were significantly lower. The flash illusion effect illustrates that hearing has an effect on vision, and whether this is an interference or facilitation effect also depends on whether the presented auditory information is consistent with the visual information. This illusion effect is not affected by the time course or spatial location and is relatively stable [51,52].

The above two effects illustrate that both synergy and conflict can occur and interfere with each other in the context of audiovisual interactions. In recent years, research on audiovisual interactions has mainly focused on the study of audiovisual synergy. Working memory, as the core component of human cognitive activity, plays a pivotal role in higher cognitive processes. Therefore, it is of great scientific value to combine the two types of studies. To this end, the present study is based on the classic research task paradigm, the oddball task (which refers to an experiment in which two or more stimuli are delivered through the same sensory channel of the subject, with one stimulus having a low probability of occurrence that ranges from approximately 10% to 30%, and the other or the remaining stimuli having a higher probability of occurrence that ranges from approximately 70% to 90%). Subjects were asked to respond only to the low probability target stimulus by pressing a button [53]. Based on this principle, a combined n-back + Go/NoGo task paradigm was created with modifications to explore the principles of working memory information processing in the context of audiovisual interactions. The research hypothesis was that, when visual and auditory stimuli are inconsistent, these stimuli will affect each other, i.e., the visual stimuli will interfere with the auditory stimuli and vice versa, according to the resource competition theory.

## 2. Materials and Methods

### 2.1. Participants

Eighty-eight undergraduate students aged 18–22 from a military medical university were recruited to participate in this study through the internet and posters. They were all male, right-handed, with normal vision and hearing, without color blindness or color weakness, and without any history of mental or neurological diseases. High intelligence was mainly measured by college entrance examination scores. Then, they were randomly divided into two groups: a single-task group of 44 people and a dual-task group of 44 people. Table 1 summarizes the sample characteristics and shows that the single-task group and dual-task group did not significantly differ in age, intelligence, or handedness (all *p* > 0.05).

All participants volunteered to participate in the experiment. Written informed consent was obtained and certain remuneration was paid after the experiment. This study was carried out in accordance with the Declaration of Helsinki and approved by the Ethics Committee for drug clinical trials of the First Affiliated Hospital of the Fourth Military Medical University.

### 2.2. Tasks and Procedure

The single-task group performed single visual n-back and auditory Go/NoGo tasks within one session. The order in which cognitive tasks were applied was pseudorandomized between participants. The dual-task group performed only the dual visual n-back + auditory Go/NoGo task as presented in the task illustrations. All the experimental programs were programmed using E-Prime3.0. The visual stimuli were presented on a 19-inch Lenovo computer monitor with a resolution of 1024 × 768, and the auditory stimuli were presented through a headset with a loudness of 70 dB. There were two kinds of sound stimulation frequencies: one was low-pitched sound with a stimulus frequency of 262 Hz. Second, the stimulus frequency was 524 Hz, a high-pitched sound, and all the participants sat 60 cm away from the computer screen. Before the experiment, the participants were first required to wear headphones, and then the instructions were presented to the participants, who were asked to carefully read them. After reading, the subjects were introduced to the experimental task content and matters needing attention according to standard procedures, and their questions were answered. When all experimental tasks began, a white fixation cross was displayed in the center of the screen for 500 ms to remind the participants that the task would start immediately, and then the judgment stimulus would be presented. The duration of the stimulus presentation was 500 ms, and the subject pressed the button or moved on to the next stimulus after 3000 ms (see Figure 1 and Figure 2).

#### 2.2.1. Single Visual Working Memory Task

The present study used the n-back task to assess visual working memory. In this task, a series of Arabic numerals was presented (0, 1, 2, 3, 4, 5, 6, 7, 8, 9), and the participants decided if the current stimulus was identical to the stimulus two trials earlier (*n* = 2 condition) or three trials earlier (*n* = 3 condition). Targets were presented in 50% of the trials. If stimuli were identical, the participants pressed “F” on the keyboard with their left hand, and if stimuli were not identical, they pressed “J” with their right hand; the task illustrations are presented in Figure 1. The key assignments were the same for all participants, and they were asked to respond quickly and accurately. There was only one block in the 2-back condition, which consisted of one run that had a duration of 5 min (80 trials). The 2-back condition started with a training run of 1 min (20 trials) with feedback indicating right or wrong reactions. When accuracy in this training stage reached 80%, the participants were considered to correctly understand the task and entered the formal experiment. Otherwise, the participants repeated the exercise. The 3-back condition was the same as the 2-back condition. The participants completed the 2-back condition first and then the 3-back condition, and the final outcomes were reaction times and accuracy of the participants in each condition.

#### 2.2.2. Single Auditory Working Memory Task

The present study used the Go/NoGo task to assess auditory working memory. In this task, the participants randomly heard one of two kinds of auditory stimuli. When hearing the auditory stimulus associated with Go trials, a high-pitched “beep” (524 Hz), the participants were instructed to press the “space” key on the keyboard to respond. When hearing the auditory stimulus associated with NoGo trials, a low-pitched “beep” (262 Hz), the appropriate response was to not press the button; the proportion of trials requiring withholding a response was 20% of the total trials, and the task illustrations are presented in Figure 2. The task included only one block of 100 trials that had a duration of approximately 6 min, and it started with a 1-min (20 trials) training phase with feedback on correct or incorrect responses. When the accuracy rate in the training phase reached 80%, the formal experiment was initiated; otherwise, the subjects had to repeat the exercise. The final outcome was the subjects’ accuracy on the NoGo trials.

#### 2.2.3. Working Memory Tasks Involving Audiovisual Interactions

The present study used a combined n-back + Go/NoGo task to assess working memory involving audiovisual interactions. In this task, the participants were presented with both a sound stimulus and a visually presented number, both of which were presented at random. The participants were asked not only to judge the sound they heard but also to remember the number presented. When the sound was a high-pitched “beep” (524 Hz), the participants were asked to judge whether the current stimulus was identical to the stimulus presented two trials earlier (*n* = 2 condition) or three trials earlier (*n* = 3 condition). Of the total trials, 40% were these target trials. If the stimuli were identical, the participants pressed “F” on the keyboard with their left hand, and if the stimuli were not identical, they pressed “J” with their right hand. However, if the sound was a low-pitched “beep” (262 Hz), the current stimulus may have been the same or different from the stimulus presented two trials earlier (*n* = 2 condition) or three trials earlier (*n* = 3 condition), but no keystroke response was required; of the total trials, 20% were these trials involving withholding responses. The keyboard response assignment was the same for all participants, and they were asked to respond quickly and accurately. There was only one block in the visual 2-back + auditory Go/NoGo condition, which consisted of one run that lasted 6 min (100 trials). The visual 2-back + auditory Go/NoGo condition started with a training run of 1 min (20 trials) with feedback indicating correct or incorrect reactions. When accuracy rate in the training stage reached 80%, the participants were considered to have an appropriate understanding of the task, and the formal experimental stage was initiated. Otherwise, the participants repeated the training. The visual 3-back + auditory Go/NoGo condition was the same as the visual 2-back + auditory Go/NoGo condition. The participants completed the visual 2-back + auditory Go/NoGo condition first and then the visual 3-back + auditory Go/NoGo condition. The dual visual n-back + auditory Go/NoGo task provided an equal quantity of target trials to the single n-back task and the single Go/NoGo task to facilitate subsequent comparisons. The final outcome was the participants’ reaction times and accuracy in the n-back condition and the accuracy on the NoGo trials.

### 2.3. Statistical Analysis

First, the data were screened to exclude data with reaction times exceeding more than 3 standard deviations from the mean so that the data for all participants conformed to a normal distribution. Second, the behavioral data were analyzed using the statistical software SPSS 22.0, mainly using independent-samples *t*-tests to compare the behavioral outcomes (accuracy, reaction time, and working memory performance) in the two groups of subjects performing the single task or the dual task. The performance related to visual working memory was calculated as the percentage of correct hits (hits on target-falsely reported nontarget), with larger values indicating better performance. The performance related to auditory working memory was inversely proportional to the NoGo commission error rate (100%-NoGo stimulus false alarm rate), with larger values indicating better performance [54]. Finally, the size of an effect (SE), *r_pb_*^2^ = *t*^2^/(*t*^2^ + *df*) was used to test the effects of audiovisual interactions on working memory. The SE was considered small when *r_pb_*^2^ = 0.010, medium when *r_pb_*^2^ = 0.059, and large when *r_pb_*^2^ = 0.138 [55]. For the above statistical analyses, the significance level was taken as α = 0.05.

## 3. Results

### 3.1. Comparisons between the Single Visual n-Back Task and Dual Visual n-Back + Auditory Go/NoGo Task

In the 2-back conditions, accuracy, reaction times, and performance data of the single-task group (visual) and dual-task group (audiovisual interaction) were statistically analyzed and are shown in Table 2.

There were no significant differences in accuracy and working memory performance between the single visual 2-back task and the dual visual 2-back + auditory Go/NoGo task (accuracy: *t*(86) = 1.465, *p* = 0.146, *r_pb_*^2^ = 0.024; performance: *t*(86) = 1.422, *p* = 0.159, *r_pb_*^2^ = 0.022) (see Figure 3a,c). In terms of reaction times, the reaction time in the dual visual 2-back + auditory Go/NoGo task was longer than that in the single visual 2-back task (916.79 ± 254.45 ms vs. 1099.445 ± 242.55 ms, *p <* 0.001, *r_pb_*^2^ = 0.14) (see Figure 3b). A correlation was computed between accuracy and reaction time in the dual visual 2-back + auditory Go/NoGo task to determine whether there were speed–accuracy trade-offs. There was a significant negative relationship [*r*(57) = −0.312, *p* < 0.05].

In the 3-back conditions, accuracy, reaction times, and performance data of the single-task group (visual) and dual-task group (audiovisual interaction) were statistically analyzed and are shown in Table 3.

We found that, with the increase in visual working memory cognitive load, accuracy and reaction times in the single visual 3-back task and dual visual 3-back + auditory Go/NoGo task showed highly significant differences (accuracy: *t*(86) = 3.750, *p* < 0.001, *r_pb_*^2^ = 0.141; reaction time: *t*(86) = −2.675, *p* < 0.001, *r_pb_*^2^ = 0.077) (see Figure 4a,b). In terms of performance, the dual visual 3-back + auditory Go/NoGo task showed poor visual working memory performance, and there was a very significant difference between the single visual 3-back task and the dual visual 3-back + auditory Go/NoGo task working memory performance [*t*(86) = 3.830, *p* < 0.001, *r_pb_*^2^ = 0.146] (see Figure 4c).

### 3.2. Comparisons between the Single Auditory Go/NoGo Task and Dual Auditory Go/NoGo + Visual n-Back Task

In different visual interference conditions, NoGo accuracy and response inhibition performance in the single-task group (auditory) and dual-task group (audiovisual interaction) were statistically analyzed and are shown in Table 4.

NoGo accuracy data in the 2-back visual interference condition are shown in Figure 5. There were significant differences in NoGo accuracy and response inhibition between the single auditory Go/NoGo task and dual auditory Go/NoGo + visual 2-back task [accuracy: *t*(86) = 2.208, *p =* 0.030, *r_pb_*^2^ = 0.053; response inhibition: *t*(86) = −2.320, *p =* 0.023, *r_pb_*^2^ = 0.059].

NoGo accuracy in the 3-back visual interference condition is shown in Figure 6. With the increase in visual cognitive interference load, NoGo accuracy in the dual auditory Go/NoGo + visual 3-back task significantly decreased compared with the single auditory Go/NoGo task [*t*(86) = 2.900, *p =* 0.005, *r_pb_*^2^ = 0.089]. In terms of response inhibition, we found a significant trend toward reduced response inhibition in the single auditory Go/NoGo task compared to the dual auditory Go/NoGo + visual n-back task (all *p* < 0.05, *r_pb_*^2^ ≥ 0.059). In particular, performance in the dual auditory Go/NoGo + visual 3-back task was significantly greater than that in the single auditory Go/NoGo task [*t*(86) = −2.774, *p* = 0.007, *r_pb_*^2^ = 0.082].

## 4. Discussion

The main purpose of our study was to explore whether audiovisual interactions (simultaneous presentation of audiovisual stimuli) interfered with or facilitated working memory to provide a theoretical basis for information processing in working memory in the context of audiovisual interactions. The n-back task is one of the most widely used tasks for studying working memory [56], as it involves complex requirements for working memory processing and engages functions related to checking updated information and storing and retaining information [57]. Meanwhile, the processing load on these memory systems can also be manipulated by controlling the size of N, allowing the investigation of the processing mechanisms of working memory under different memory loads [58,59]. The Go/NoGo task is the most classic research paradigm for evaluating inhibitory interference in working memory [60,61]. Therefore, the n-back task was mainly used to measure the visual working memory of the subjects, while the Go/NoGo task was used to measure the auditory working memory of the subjects.

The results showed that, in the visual 2-back condition, auditory working memory accuracy was reduced in the context of an audiovisual interaction task compared with the single auditory working memory task, supporting the theory of resource competition with audiovisual interactions. When visual and auditory stimuli are inconsistent, there is conflict between the two that results in competition for resources, which impedes the brain’s response [62]. On the other hand, inhibitory performance (i.e., the inverse of the NoGo commission error rate, thus having better performance, as was indicated by higher values) in the audiovisual working memory task was higher than that in the single auditory working memory task. These results suggest that our visual perception influences our auditory perception in conditions involving audiovisual interactions, and the participants needed to engage more attention resources to suppress the interference associated with the irrelevant visual information; this resulted in response inhibition in audiovisual interaction conditions being better than that in auditory alone conditions. However, in the context with interference from the auditory Go/NoGo task, we found that there was no difference in accuracy between the 2-back task involving audiovisual interactions and the single visual 2-back task. However, the reaction time in the 2-back task involving audiovisual interactions was significantly longer than that in the single 2-back task, and there was a trade-off between speed and accuracy; this finding indirectly indicated that the participants improved accuracy by sacrificing reaction time, which was a reflection of a processing strategy. This is consistent with the results of previous studies on spatial working memory in the context of the simultaneous presentation of audiovisual stimuli. Compared with pure visual or auditory stimuli, there was no difference in working memory accuracy in the context of simultaneously presented audiovisual stimuli despite the delay of reaction time [63]. Finally, in terms of visual working memory performance, we found no difference between 2-back task performance with audiovisual interactions and 2-back task performance with visual stimuli alone, indicating that hearing did not affect vision, which is completely inconsistent with previous research results evaluating audiovisual interactions. Neither the theory of collaboration nor the theory of resource competition in the context of audiovisual interactions is supported. The authors speculate that the reason may be related to the cognitive processing load on our working memory. Researchers have found that the 1-back condition is a pure memory retention task that mainly measures the short-term memory of the subjects, while the 3-back condition is the maximum memory load condition to which subjects can correctly respond in most cases [64,65], consistent with the performance being limited by a working memory capacity that can maintain up to four objects at the same time [24,66]. Therefore, the 3-back condition involves working memory under high load, while the 2-back condition between the 1-back and 3-back conditions involves working memory under low load. However, our cognitive processing load is limited by our working memory processing load, which is positively correlated with the allocation of our attention resources [67]. In the case of effective allocation and control of cognitive resources, cognitive load is negatively correlated with cognitive resource reserve [68]. That is, the lower the perceived cognitive load of an individual, the more sufficient our cognitive resource reserve will be so that we can effectively restrain interference by irrelevant stimuli. At the same time, the college students in our study belong to a particular group of people with higher cognitive abilities. In the 2-back (low cognitive load) conditions, there is most likely a ceiling effect, resulting in the stimulus having acoustic interference, but the single visual stimulus and audiovisual stimulus in the visual working memory task showing no difference in performance.

In conclusion, in the 2-back (low cognitive load) condition, the presence of the visual stimulus interfered with the auditory working memory task, while the presence of auditory stimulus did not affect the visual working memory task. The authors speculated that these effects were mainly related to the cognitive load on working memory. Therefore, the following is an exploration of whether these findings were related to our conjecture and further explore whether the influence of audiovisual interactions on working memory task represents interference or facilitation. We found that, with interference in the visual 3-back (high cognitive load) task, accuracy of auditory working memory decreased compared to accuracy in a single auditory working memory task, which is consistent with previous studies. In the context of audiovisual interactions, the auditory stimulus was used as the target stimulus, and the inconsistent visual stimulus was used as the interference stimulus, which will affect the subjects’ brain response to a certain extent [62]. The degree of this effect depends on the load imposed by the inconsistent visual interference stimulus; that is, the interference effect will increase with increases in visual cognitive load [25]. Indeed, our results showed that accuracy of auditory working memory in the visual 3-back condition was significantly lower than that in the visual 2-back condition. This is consistent with recent findings in spatial conflict tasks with audiovisual interactions. Compared with the audiovisual stimuli presented in a consistent position, we found that the interaction with audiovisual stimuli presented in an inconsistent position resulted in recognition rates of the auditory target stimuli were related to our visual cognitive processing load; that is, the more complex the visual cognitive processing task was, the significantly lower the recognition rate of the auditory target stimulus [69]. This strongly suggested that, when the processing information provided by visual and auditory stimuli is inconsistent, our vision eventually affects our hearing, in both visual 2-back and visual 3-back conditions, which supports the theory of resource competition in the context of audiovisual interactions. Meanwhile, inhibitory performance in the auditory working memory task continued to increase with the audiovisual interaction, which further indicated that the participants’ visual 3-back (high cognitive load) interference had a greater impact on auditory working memory. Compared with visual 2-back (low cognitive load) conditions, visual 3-back (high cognitive load) conditions have a greater need for response inhibition to suppress irrelevant stimuli, which facilitate the response inhibition effect in auditory working memory with audiovisual interaction from small to medium [55].

However, in the case of auditory Go/NoGo task interference, compared with the single visual 3-back condition, accuracy in the 3-back condition with audiovisual interaction significantly decreased, and response processing times were the longest, which fully indicated that the auditory stimuli interfered with the information resolution of visual working memory at this time. Our vision being influenced by auditory stimuli [48,70] supports the audiovisual resource competition theory in conditions involving interactions, consistent with findings with audio and visual stimulation. The brain does not match the processing of audio and visual stimulation, as there are certain processing difficulties, thus indirectly impacting the corresponding reaction time [71], causing the participants to react more slowly. Accuracy was lower, and there was no trade-off between speed and accuracy [72,73,74]. Finally, we found that the working memory performance in the 3-back condition with the audiovisual interaction was significantly decreased compared to performance in the 3-back condition with visual stimuli alone. This is consistent with previous research results. With the integration of the stop-signal task with response inhibition performance in the n-back task of working memory, working memory performance will decline, which is explained by the combination of the requirements of the n-back and stop-signal tasks. The subjects need to remember more response rules. Both of these tasks are presented through visual channels, and the presence of a visual stop-signal task interferes with the processing of visual working memory, resulting in a decline in visual working memory performance [75]. However, some researchers hold the opposite view; unlike visual noise stimulation, auditory interference in conditions of high load can be further processed, and the interference in high-load conditions causes an effect [76]. To achieve visual cognitive load conditions, auditory interference stimulates the maximum processing intensity [77]. This explanation may be closer to our results. Compared with the audiovisual interaction in the 2-back (low cognitive load) conditions, the presence of auditory stimuli in the audiovisual stimuli in the 3-back (high cognitive load) condition has a large effect on visual working memory performance [55]. Most importantly, both the visual working memory n-back task and the auditory working memory Go/NoGo task are associated with the central executive system of working memory. Our central executive system is a control system with limited attention. When participants switch their attention from one modal item (visual working memory task) to another modal item (auditory working memory task), that is, there is a constant switching between visual and auditory working memory tasks, such switching requires some effort and increases the cognitive load [78]. However, the original cognitive load on working memory is already the load to which the subjects can make the maximum response, which is bound to greatly affect performance in working memory tasks [79,80]. This is consistent with our findings that 3-back working memory performance was significantly decreased in both visual 3-back working memory performance and performance in the context of audiovisual interactions. Therefore, it was preliminarily shown that the influence of audiovisual interactions on working memory tasks was related to cognitive load, which was consistent with the authors’ prediction.

In summary, this study of the effects of audiovisual interactions in the context of working memory tasks confirms an interesting previous finding that visual dominance is predominant during engagement in tasks. Previous studies have shown that, when faced with stimuli from different sensory modes, humans usually choose the most accurate processing mode according to the task requirements [81]. Existing research findings and daily life experiences have confirmed that vision is typically the main sensory mode of humans in many cases, although under some conditions, we may use the auditory or somatosensory system if it is advantageous [82]. However, in most cases, humans show a strong tendency to rely on visual information rather than on other forms of sensory information [83], and our results support this finding. That is, our vision always interferes with our auditory processing, and our auditory does not always interfere with our visual processing. The presence of auditory stimuli greatly interferes with visual processing only when it is engaged in high-load cognitive tasks. This is because, when a large number of sensory pathways are present at the same time, our brain does not give equal weight to each sensory pathway. Visual and auditory channels are the two main information-processing channels for humans, and among all the channels, 83% of information comes from visual sensory channels and 11% from auditory sensory channels [84]. Therefore, when our brain processes information, the allocation of visual resources must be greater than that of auditory resources, resulting in prioritizing processing of visual stimuli and guiding other sensory channels [85]. A large number of studies have shown that visual stimuli can dominate auditory stimuli in participants completing spatial judgment tasks. For example, visual stimuli can adjust the perception of auditory stimuli [86], and visual stimuli can influence our judgment of the location of obvious auditory stimuli [42]. In subjects completing the shape judgment task, the visual channel has been shown to dominate the tactile channel [87]. Additionally, visual stimuli have been reported to dominate somatosensory perception in subjects attempting to judge the position of limbs in space [37]. Ward et al. [88] have suggested that visual stimulation is superior to auditory stimulation in spatial positioning tasks because the visual input topology is mapped in the retina, and this process is continuous and applies to higher visual processing stages. Therefore, when there is a conflict between visual information and auditory information, the visual channel predominates so that the problem can be solved. Thus, visual perception has a higher priority in the integration process of the whole sensory system, that is, processing mechanisms dominated by visual stimuli are adopted [89]. The most classic dominant effect of visual stimulus is the Colavita effect, which refers to the phenomenon wherein people responding to a visual stimulus presented at the same time as an auditory stimulus often cannot respond to the auditory stimulus [90]. This phenomenon is now commonly referred to as the Colavita visual dominance effect [91], which means that our visual processing always disrupts our auditory processing, and this disruption is often considered unconscious and essentially undetected [90]. This is in agreement with our experimental results. Our study investigated the influence of audiovisual interactions on working memory and found that our vision will always disrupt our auditory processing. This kind of interference depends on the cognitive load associated with the visual stimulation; that is, given the cognitive load of visual stimulation, the greater the auditory stimulation, the greater the degree of the interference on working memory. At the same time, we also found that auditory interference of visual stimulus processing is conditionally limited, and only when visual stimuli are engaged in processing high-load cognitive tasks will the presence of auditory stimuli interfere with the visual information processing, and the advantage of auditory interference is more obvious [92]. This may be due to the limited processing resources of the central executive system of our working memory, leading to the processing advantages of sensory channels first shifting to modes involving these processing resources [93]. Therefore, due to the increased cognitive load on working memory, subjects have to pay attention to and use a large number of cognitive resources to deal with a current complex cognitive task, which leaves insufficient cognitive resources to suppress irrelevant stimuli. At this point, when task-unrelated sound stimuli appear with a small probability, these stimuli can still capture the subjects’ attention and cause an increase in nonrandom attention shifts, i.e., interference effects where our auditory experience interferes with our visual experience [94]. However, when processing in working memory tasks with low cognitive load, we have enough cognitive resources to suppress irrelevant sound stimuli; that is, the presence of auditory stimuli does not interfere with our visual working memory processing.

### Limitations

Our study presents some limitations. First of all, the interaction between vision and hearing was greater under high load 3-back than under low load 2-back. Since this experiment is an exploratory study, the authors speculate that this is an interference caused by increased cognitive load. Therefore, in the following studies, we hope to further explore the causes of cognitive load itself or the paradigm attributes of combined tasks. In addition, the current studies are all behavioral studies, lacking evidence of some physiological indicators. Therefore, we hope to add some physiological indicators in future experimental studies to provide strong evidence for our experimental results.

## 5. Conclusions

Our results support the idea that the effects of audiovisual interactions on working memory tasks are related to cognitive load and are dominated by competition for visual stimuli. Compared with the impact in a single auditory working memory task, the competitive effects of audiovisual interactions were more pronounced as the visual cognitive load increased. Compared with the impact in a single visual working memory task, the stimuli in the audiovisual interaction competed with each other and produced additional interference only when the visual cognitive load was higher; at lower visual cognitive loads, the two stimuli were no longer in competition and neither facilitated nor interfered, resulting in consistent speed and accuracy, which is consistent with a trade-off between speed and accuracy.

## Figures and Tables

**Figure 1 brainsci-12-00886-f001:**
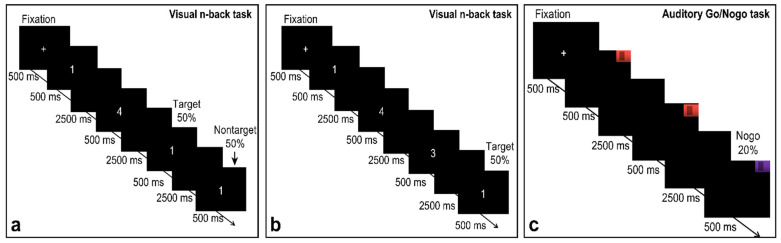
Single-task schematic. (**a**) represents the single visual working memory task in the 2-back condition, (**b**) represents the single visual working memory task in the 3-back condition, and (**c**) represents a single auditory working memory task.

**Figure 2 brainsci-12-00886-f002:**
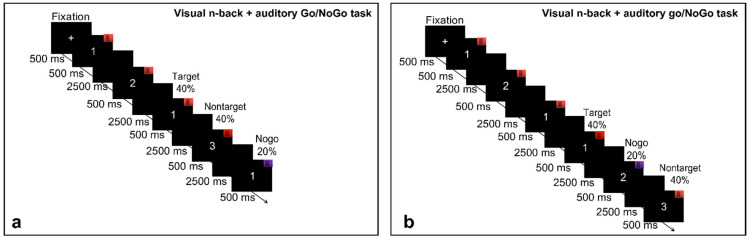
Dual-task schematic. (**a**) represents the audiovisual interaction working memory task in the 2-back condition, and (**b**) represents the audiovisual interaction working memory task in the 3-back condition.

**Figure 3 brainsci-12-00886-f003:**
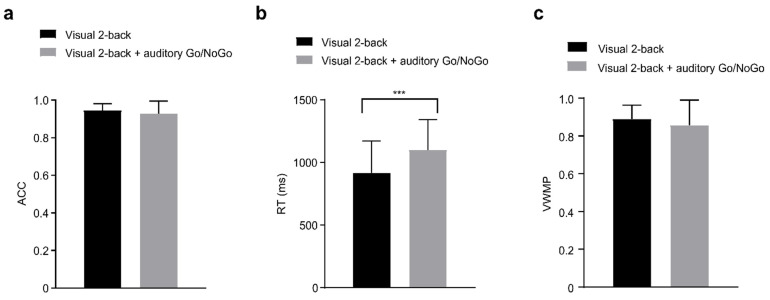
In the 2-back conditions, (**a**) single-task (visual) and dual-task (audiovisual interaction) groups for accuracy (ACC); (**b**) single-task (visual) and dual-task (audiovisual interaction) groups for reaction time (RT); (**c**) as well as single-task (visual) and dual-task (audiovisual interaction) groups for performance data (visual working memory performance, VWMP). Between the single-task (visual) and dual-task (audiovisual interaction) groups revealed that the RT in the dual visual 2-back + auditory Go/NoGo task was significantly longer than that in the single visual 2-back task. All the data are presented as the mean ± S.D (*n* = 44); *** *p* < 0.001.

**Figure 4 brainsci-12-00886-f004:**
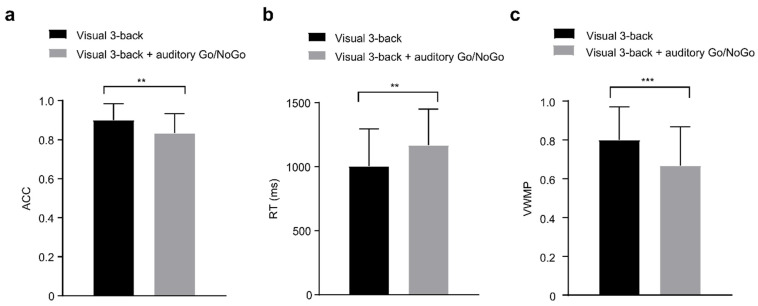
In the 3-back conditions, (**a**) single-task (visual) and dual-task (audiovisual interaction) groups for accuracy (ACC); (**b**) single-task (visual) and dual-task (audiovisual interaction) groups for reaction time (RT); (**c**) as well as single-task (visual) and dual-task (audiovisual interaction) groups for performance data (visual working memory performance, VWMP). There were differences in ACC and RT as well as VWMP between the single-task group (visual) and the dual-task (audiovisual interaction) group. All the data are presented as the mean ± S.D (*n* = 44); ** *p* < 0.01, *** *p* < 0.001.

**Figure 5 brainsci-12-00886-f005:**
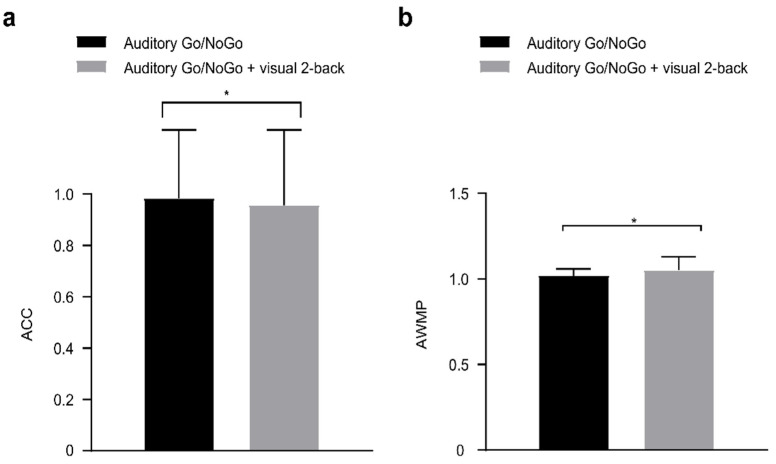
In the visual 2-back interference condition, (**a**) single-task (auditory) and dual-task (audiovisual interaction) groups for accuracy (ACC); (**b**) single-task (auditory) and du-al-task (audiovisual interaction) groups for performance data (auditory working memory performance, AWMP). Between the single auditory Go/NoGo task and the dual auditory Go/NoGo + visual n-back task revealed significant differences, *p* < 0.05. All the data are presented as the mean ± S.D (*n* = 44); * *p* < 0.05.

**Figure 6 brainsci-12-00886-f006:**
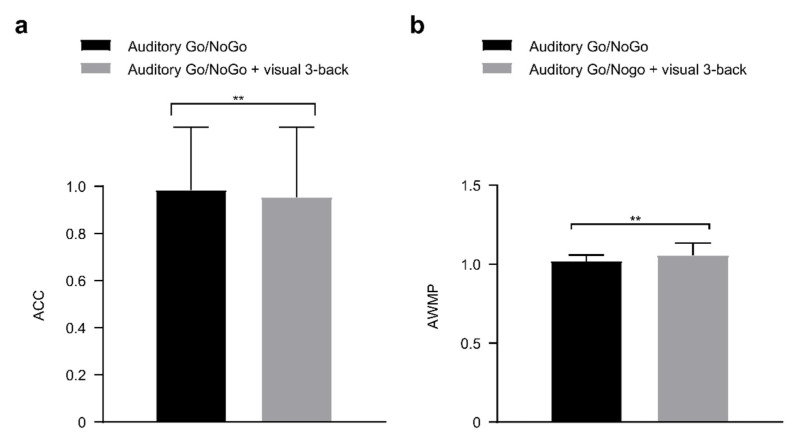
In visual 3-back interference conditions, (**a**) single-task (auditory) and dual-task (audiovisual interaction) groups for accuracy (ACC); (**b**) single-task (auditory) and dual-task (audiovisual interaction) groups for performance data (auditory working memory performance, AWMP). Between the single auditory Go/NoGo task and the dual auditory Go/NoGo + visual n-back task groups revealed significant differences, *p* < 0.01. All the data are presented as the mean ± S.D (*n* = 44); ** *p* < 0.01.

**Table 1 brainsci-12-00886-t001:** Sample characteristics of the single-task group and dual-task group, mean ± standard deviation, Cohen’s effect size *r_pb_*^2^, *t*, and *p*-values are presented.

	Single-Task Group	Dual-Task Group	*t*	*p*	*r_pb_* ^2^
*n*	44	44	-	-	-
Age in years	20.05 ± 1.08	19.75 ± 1.04	1.311	0.193	0.01
IQ	598.66 ± 30.86	606.25 ± 24.889	−1.270	0.207	0.02

**Table 2 brainsci-12-00886-t002:** Behavioral performance parameters in the single-task group (visual) and dual-task group (audiovisual interaction) were statistically analyzed (mean ± standard deviation).

Task Type	Memory Condition	Behavioral Results		
		Accuracy	Reaction Time	Visual Performance
Single-task group (visual)	2-back	0.94 ± 0.04	916.79 ± 254.45	0.89 ± 0.07
Dual-task group (audiovisual interaction)	2-back	0.94 ± 0.06	1099.445 ± 242.55	0.86 ± 0.13

**Table 3 brainsci-12-00886-t003:** Behavioral performance parameters in the single-task group (visual) and dual-task group (audiovisual interaction) were statistically analyzed (mean ± standard deviation).

Task Type	Memory Condition	Behavioral Results		
		Accuracy	Reaction Time	Visual Performance
Single-task group (visual)	3-back	0.90 ± 0.08	1004.19 ± 290.54	0.79 ± 0.18
Dual-task group (audiovisual interaction)	3-back	0.83 ± 0.06	1167.97 ± 282.59	0.66 ± 0.19

**Table 4 brainsci-12-00886-t004:** Behavioral performance parameters in the single-task group (auditory) and dual-task group (audiovisual interaction) were statistically analyzed (mean ± standard deviation).

Different Visual Interference Conditions	Task Type	Behavioral Results	
		Accuracy	Auditory Performance
	Single-task group (auditory)	0.98 ± 0.04	1.02 ± 0.04
2-back	Dual-task group (audiovisual interaction)	0.96 ± 0.05	1.05 ± 0.08
3-back	Dual-task group (audiovisual interaction)	0.95 ± 0.06	1.06 ± 0.08

## Data Availability

The datasets presented in this article are not readily available be- cause the datasets involve unfinished research projects. If necessary, requests to access the datasets should contact the corresponding author.

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
