# Peer review of "Effects of Audiovisual Interactions on Working Memory Task Performance—Interference or Facilitation"

_brainsci, 2022, doi:10.3390/brainsci12070886_

Round 1
Reviewer 1 Report
The study “Effects of Audiovisual Interactions on Working memory task performance – Interference or Facilitation” applied a combination of n-back and go/nogo tasks in the auditory and visual sensory modalities to investigate the effects of task load between the senses. Load was manipulated by applying 2-back or 3-back conditions in unisensory and multisensory (AV) conditions. For the low load conditions auditory stimuli had no effect on visual WM and visual stimuli purportedly had a small effect on auditory WM. However, for the high load conditions auditory stimuli had a large effect on visual WM while visual stimuli purportedly had a medium effect on auditory WM.
General comments:
It is not quite clear why a combination of tasks were applied. Why were n-back not used for both auditory and visual modalities? It is hard to draw the conclusions you draw, since you are comparing different tasks (WM and Response Inhibition) in some of the comparisons. The go/nogo task does not measure working memory as proclaimed. It measures the ability to inhibit responses, so these can’t be compared directly. Also, it is not sure if these results stem from audiovisual interactions as described or just from increased task load. The experimental design does not answer this.
Cognitive/attentional load is first introduced in the discussion, while this is quite relevant to the thesis of this paper. Please introduce this properly in the introduction.
Itemized comments:
P. 1 – The title has uneven capitalization.
P. 1 – The abstract is too detailed. No need to include the number and age of subjects here.
P. 1, L. 16 – the semicolon looks awkward here.
P. 1, L. 39 – Put “that includes…visuospatial sketchpad” in brackets for readability
P. 2, L. 61 – a multidisciplinary focus
P. 2, L. 64 – between (not among)
P. 2, L. 65– simultaneously (instead of at the same time)
P. 2, L. 69-75 - This seems a but handwavy but certainly these are the most used sensory modalities. Maybe make the point of visual dominance here that is introduced in the discussion
P. 2, L. 77-97 – The Temporal order Judgment literature between the senses may be relevant here e.g.
Zampini, M., Shore, D. I., & Spence, C. (2005). Audiovisual prior entry. Neuroscience Letters, 381, 217–222.
Vibell, J., Klinge, C., Zampini, M., Spence, C., & Nobre, A.C. (2007). Temporal order is coded temporally in the brain: Early ERP latency shifts underlying prior entry in a crossmodal temporal order judgment task. Journal of Cognitive Neuroscience, 19, 109-120.
P. 3, L. 113 – The above two effects illustrate that (many ppl found that before you)
P. 3, L. 119 – Remove ‘involving audiovisual interactions’
P. 3, L. 127 – don’t forget to add your hypothesis
P. 4, L. 150-151 – Stimuli were (plural x2)
P. 6, L. 239 – Go/nogo task does not measure working memory and should not be directly compared as such
P. 7, L. 275 – highly significant (not very)
P. 10, L. 318 – Not sure if its from audiovisual interactions or just increased task load
P. 10, L. 331 - ‘Results showed’ instead of ‘according to our results we showed that’
P. 10, L. 336 – clarify ‘inhibitory performance’
P. 10, L. 338 – ‘suggest’ instead of ‘fully indicate’, btw we already know fully well that vision and audition influence each other.
P. 12, L. 448 – rephrase ‘fully demonstrated’
P. 12, L. 452-454 – you did not discover this. Its called visual dominance and is a broad set of literature that probably should be discussed more fully
Sinnett, S., Spence, C. & Soto-Faraco, S. Visual dominance and attention: The Colavita effect revisited. Perception & Psychophysics 69, 673–686 (2007). https://doi.org/10.3758/BF03193770
P. 12, L. 458-460 – I am not sure I agree with these numbers as its difficult to quantify exactly, but they should at least be referenced
P. 12, L. 460-470 – This restates what you already said in the introduction.
Author Response
Dear editors and reviewers,
Thank you for your letter and for the reviewers’ comments concerning our manuscript entitled “Effects of Audiovisual Interactions on Working Memory Task Performance—Interference or Facilitation” (ID: brainsci-1756218). Those comments are all valuable and very helpful for revising and improving our paper, as well as the important guiding significance to our researches. We have studied comments carefully and have made correction which we hope meet with approval. At the same time, according to your request, we have downloaded a copy of the original manuscript with all the changes marked in red by using the track changes mode in MS Word.
We tried our best to improve the manuscript and made some modifications to it. These changes will not affect the content and framework of the paper. Here, we do not list the changes, but they are marked in red in the revised document.
We sincerely thank the editors and reviewers for their enthusiastic work and hope that the correction will be approved. More importantly, we hope that the revised manuscript will be accepted for publication by the Journal of Brain Sciences.
Finally, please refer to Attachment 1 for the revised version, see page 20-30 of Attachment 1 for the reviewer's comments, and see page 31 of Attachment 1 for the polishing certificate.
Once again, thank you very much for your comments and suggestions.
Thank you and best wishes
Sincerely yours
Yang He

Reviewer 2 Report
The work studies the effects of audiovisual interactions on working memory tasks. Results indicate that these are related to cognitive load and are dominated by competition for visual stimuli. I am not an expert in this topic, but the methods and results look valid to me.
I suggest to incorporate a native speaker for proof reading the manuscript (or make use of a professional proof reading / editing service).
The references need also to be checked carefully, for example:
76. 76. Kimura, M.; Katayama, J.; Murohashi, H. Underlying mechanisms of P3a-task-difficulty effect. Psychophysiology. 2008, 45,
77. 731-741; DOI:10.1111/j.1469-8986.2008.00684.x.
Author Response
Dear editors and reviewers,
Thank you for your letter and for the reviewers’ comments concerning our manuscript entitled “Effects of Audiovisual Interactions on Working Memory Task Performance—Interference or Facilitation” (ID: brainsci-1756218). Those comments are all valuable and very helpful for revising and improving our paper, as well as the important guiding significance to our researches. We have studied comments carefully and have made correction which we hope meet with approval. At the same time, according to your request, we have downloaded a copy of the original manuscript with all the changes marked in red by using the track changes mode in MS Word.
We tried our best to improve the manuscript and made some modifications to it. These changes will not affect the content and framework of the paper. Here, we do not list the changes, but they are marked in red in the revised document.
We sincerely thank the editors and reviewers for their enthusiastic work and hope that the correction will be approved. More importantly, we hope that the revised manuscript will be accepted for publication by the Journal of Brain Sciences.
Finally, please refer to attachment 1 for the revised version, and the polishing certificate is on page 21 of attachment 1.
Once again, thank you very much for your comments and suggestions.
Thank you and best wishes
Sincerely yours
Yang He
Thank you and best wishes
Sincerely yours
Yang He
Response to Reviewer2 Comments
Point 1: I suggest to incorporate a native speaker for proof reading the manuscript (or make use of a professional proof reading / editing service).
Response 1: First of all, thank you for your highly recognition of this article written by our team. We are very honored and thank you again. Second, I would like to sincerely apologize to you for the poor language of our manuscript. We worked on the manuscript for a long time and the repeated addition and removal of sentences and sections obviously led to poor readability. We have now worked on both language and readability and have also involved native English speakers for language corrections (See the polishing certificate is on page 21 of attachment 1). We really hope that the flow and language level have been substantially improved.
Point 2: The references need also to be checked carefully, for example:
- 76. Kimura, M.; Katayama, J.; Murohashi, H. Underlying mechanisms of P3a-task-difficulty effect. Psychophysiology. 2008, 45,
- 731-741; DOI:10.1111/j.1469-8986.2008.00684.x.
Response 1: Thank you very much for your reminding. We are deeply sorry for the negligence of the original manuscript reference. According to your reminding, we carefully checked all the references cited in this article, and all the references with inappropriate formats were changed and marked in red (See attachment 1 for reference).
For example:(99. Kimura, M.; Katayama, J.; Murohashi, H. Underlying mechanisms of P3a-task-difficulty effect. Psychophysiology. 2008, 45, 731-741; DOI:10.1111/j.1469-8986.2008.00684. x.)
Finally, once again, thank you very much for your comments and suggestions. and I hope to learn more from you.

Round 2
Reviewer 1 Report
Thank you for addressing my all my comments thoughtfully. I am happy with all the comments. The two issues of comparing quite different tasks and quite different task loads remain. Those are fundamental design flaws that should be addressed in future studies. I am OK with that you just mention them in this study, but believe, in general, that they are important to address to really get to the bone on this topic.